# A Comprehensive Review on Combinatorial Film via High-Throughput Techniques

**DOI:** 10.3390/ma16206696

**Published:** 2023-10-15

**Authors:** Dongxin Wang, Wei Jiang, Shurong Li, Xuehui Yan, Shuaishuai Wu, Haochen Qiu, Shengli Guo, Baohong Zhu

**Affiliations:** 1State Key Laboratory of Special Rare Metal Materials, Northwest Rare Metal Materials Research Institute Ningxia Co., Ltd., Shizuishan 753000, China; wangdongxin123@126.com; 2State Key Laboratory of Huazhong University of Science and Technology Material Forming and Die Technology, Huazhong University of Science and Technology, Wuhan 430074, China; 3GRIMAT Engineering Institute Co., Ltd., Beijing 101407, China; wjiang111@126.com (W.J.); yanxuehui@grinm.com (X.Y.); wushuaishuai@grinm.com (S.W.); qiuhaochen@grinm.com (H.Q.); guoshengli@grinm.com (S.G.); 4China GRINM Group Corporation Limited, Beijing 100088, China

**Keywords:** high-throughput, gradient, deposition, film

## Abstract

Numerous technological advancements in the 21st century depend on the creation of novel materials possessing enhanced properties; there is a growing reliance on materials that can be optimized to serve multiple functions. To efficiently save time and meet the requirements of diverse applications, high-throughput and combinatorial approaches are increasingly employed to explore and design superior materials. Among them, gradient thin-film deposition is one of the most mature and widely used technologies for high-throughput preparation of material libraries. This review summarizes recent progress in gradient thin-film deposition fabricated by magnetron sputtering, multi-arc ion plating, e-beam evaporation, additive manufacturing, and chemical bath deposition, providing readers with a fundamental understanding of this research field. First, high-throughput synthesis methods for gradient thin films are emphasized. Subsequently, we present the characteristics of combinatorial films, including microstructure, oxidation, corrosion tests, and mechanical properties. Next, the screening methods employed for evaluating these properties are discussed. Furthermore, we delve into the limitations of high-throughput preparation and characterization techniques for combinatorial films. Finally, we provide a summary and offer our perspectives.

## 1. Introduction

Materials play a fundamental role in advancing society and are indicative of our progress and achievements as a civilization. The pace of progress in all areas is heavily reliant on the continuous development of materials, making it a vital component in promoting economic growth and ensuring national security [1,2]. However, in current global scientific and technological revolution and industrial transformation, one of the largest bottlenecks is the lack of adequate material technology. The conventional trial-and-error approaches for materials research and development are resource- and time-consuming. Hence, it has become imperative to develop an efficient and systematic method for studying structural alloys with targeted properties [3,4,5,6,7,8,9,10,11,12].

At the start of the 21st century, Ceder, a researcher at the Massachusetts Institute of Technology (MIT) in Cambridge, MA, USA, was influenced by high-throughput, data-driven methods for discovering new materials and drew inspiration from the Human Genome Project. He pondered whether material scientists could learn from the experiences of geneticists. In 2006, Ceder founded the Materials Genomics Project at MIT, utilizing an enhanced data-mining algorithm to forecast lithium-based materials for use in electric vehicle batteries. By 2010, the project had expanded to include approximately 20,000 anticipated compounds. Concurrently, at Duke University in Durham, NC, USA, Curtarolo established the Materials Genomics Center, which concentrated on exploring metal alloys [2,13,14,15].

In 2011, the United States officially launched the Materials Genome Initiative (MGI) to enhance the United States’ global competitiveness, establishing a new materials research and development model for future-oriented integrated computing, experiments, and databases. The European Union has also put forward a series of policies, such as the New Material Discovery NOMAD plan, Germany’s Industry 4.0 strategy, Russia’s Materials and Technology Development Strategy before 2030, and China’s Materials Genetic Engineering, which apply the exploration of new materials, innovative design and research, and development of materials as the primary development goals [1,2,16,17,18,19,20,21,22,23,24,25,26].

The establishment of the MGI has become a turning point in data-driven materials science, and the database has gradually evolved into a data center that provides materials data and fundamental analysis services. As the fourth paradigm of data-driven scientific development, materials genetic engineering combines high-throughput computing and design, high-throughput preparation, high-throughput characterization, material databases, and artificial intelligence, significantly shortening the material research and development cycle and reducing the research and development cost to rapidly develop new materials and meet the growing performance requirements [27,28,29,30,31,32,33,34,35,36,37,38]. The objective and purpose of MGI include pioneering a fresh approach to research and development (R&D) and a novel technological framework that seamlessly integrates and fosters collaborative innovation across the spectrum of materials R&D, manufacturing, and application. This initiative aims to enhance R&D productivity substantially, facilitate the adoption of engineering applications, and address the growing and pressing need for new materials in economic and social progress [16,17,18].

High-throughput preparation and characterization have been proposed for the large-scale screening of alloy compositions in the design of advanced alloys. High-throughput approaches, as their name implies, provide an alternative to conducting experiments one at a time or through single-step processes. Instead, they enable researchers to carry out multiple investigations within a relatively short timeframe. This is typically achieved through swift sequential automation or concurrent measurements, involving myriad material variables. Consequently, the application of high-throughput experiments serves the purpose of swiftly screening materials; the insights gained from these experiments serve as a foundation for subsequent, more detailed investigations [19,20,21,22,23,24,25,26].

Composition spread alloy films (CSAFs), first invented in the field of chemistry, are one of the most mature and widely used technologies for high-throughput preparation of material libraries [11,12]. Indeed, many valuable discoveries in materials science have been made due to adopting CSAF approaches, particularly in corrosion, oxidation, magnetic, and mechanical properties.

Several review articles have focused on the MGI, providing valuable information regarding its current situation and development. However, few reviews have comprehensively presented the fabrication, screening methods, microstructures, properties, and challenges of CSAF approaches. Therefore, this review focuses on applying combinatorial film high-throughput preparation and characterization techniques. The progress, application fields, and common challenges associated with this area are also within the scope of the review.

Section 2 introduces typical high-throughput synthesizing methods, such as magnetron co-sputtering, multi-arc ion plating, e-beam evaporation, additive manufacturing, chemical bath deposition, etc. The essence of high-throughput synthesis methods lies in the creation of CSAFs. The preparation methods discussed in this paper are well-developed and extensively employed. Section 3 focuses on high-throughput characterization methods for screening the microstructure and properties, like SEM, TEM, XPS, Raman, etc. Section 4 introduces the screen methods on oxidation, corrosion, and mechanical properties. Finally, we present an outlook on the future research directions and challenges of developing high-throughput techniques.

Moreover, the literature on high-entropy alloys (HEA) is vast. However, our work focuses on introducing the two types of HEAS: 3D transition element HEA and refractory element HEA. In addition, the study of oxidation and corrosion performance is aimed at alloys subjected to long-term or harsh environmental conditions. These materials must withstand such degradation over an extended period, and high-throughput techniques enable rapid analysis, saving significant time and effort.

## 2. High-Throughput Synthesis Methods

The purpose of high-throughput synthesis is to obtain CSAFs. To date, combinatorial film high-throughput synthesis techniques, such as magnetron co-sputtering, multi-arc ion plating, e-beam evaporation, electrodeposition, and additive manufacturing, utilize the generation of compositional gradients to achieve a wide range of alloy compositions.

### 2.1. Magnetron Co-Sputtering

Magnetron co-sputtering has been widely used for high-throughput preparation, and can be used to prepare CSAFs of various materials. In addition, the equipment can be easily acquired, facilitating the rapid proliferation of its extensive utilization. The process involves loading constituent elements into sputtering guns, which are then utilized as target materials for the confocal, magnetron co-sputtering of thin films exhibiting compositional gradients onto a substrate. Compositional gradients are generated by selectively loading one or more elemental materials into each sputtering gun in predetermined ratios. The resulting film is obtained without substrate rotation. The composition range of the film can be controlled by adjusting the target–substrate angle and target power. Hence, the element content increases as power increases or as the target substrate angle changes in the direction of the substrate. A schematic representation of magnetron sputtering utilized for producing a compositionally varied Nb-Si-based alloy film is presented in Figure 1 [25].

The advantages of magnetron co-sputtering are [40,41,42,43,44]:(a)Wide composition range. By adjusting the deposition angle and power of the targets, the CSAFs can vary over a large elemental composition range.(b)High-quality. The thin films prepared by magnetron co-sputtering exhibit low defect density and uniform thickness.(c)Wide applicability. Magnetron co-sputtering can prepare thin films of various materials, including metals, semiconductors, and insulators, and has a wide range of applications.(d)Environmentally friendly. Preparing the CSAFs by magnetron co-sputtering does not produce toxic or hazardous waste or pollutants, making it relatively environmentally friendly.

However, magnetron co-sputtering requires a significant amount of energy to excite the surface atoms or molecules of the target material for sputtering. Therefore, the efficiency is low, generally taking several hours, and the thickness of fabricated films is relatively thin.

### 2.2. Multi-Arc Ion Plating

Another common high-throughput preparation technique is multi-arc ion plating. The compositional gradient films are deposited in an ultra-high vacuum chamber by multi-arc ion plating. To produce compositional gradients, two targets with different compositions are simultaneously employed to obtain the films. Two targets are positioned within the target holder, one in the upper section and the other in the lower section; the substrates are then placed in front of the targets. A schematic image of multi-arc ion plating for generating Nb-Si based alloy film is shown in Figure 2 [45]. The Nb-Si based target was placed in the upper portion of the target holder and the pure chromium target was placed in the lower portion; the substrates were placed in front of the targets. The atoms of the target can be deposited onto the substrate at high temperatures. From the Nb-Si based alloy target end to the Cr target end, the concentration of Cr gradually increases, while the concentrations of Nb, Si, and Ti gradually decrease.

The advantages include [46,47,48,49,50,51]:(a)High deposition rate. Compared with magnetron sputtering, multi-arc ion plating has a higher deposition rate, which allows for the preparation of relatively thick films in a short time, thus increasing production efficiency.(b)High adhesion. The adhesion strength between the film and the substrate is high due to the ion beam bombardment-induced creation of a strong interfacial bond.

However, certain disadvantages are also apparent:(a)Narrow composition range. Due to the fixed deposition angle of the targets, the composition range of the CSAFs is constrained. To broaden the range, it is necessary to increase the substrate area, which results in an associated increase in experimental costs.(b)Low-quality. Multi-arc plating films are typically characterized by microparticles formed when the droplets emitted from the arc spots of the targets solidify.

### 2.3. E-Beam Evaporation

Electron beam physical vapor deposition (EB-PVD) is an alternative method for fabricating CSAFs. This method uses a high-energy electron beam to heat the target material at a specific spot, causing it to melt and sublimate. It is then deposited on the surface of the substrate in molecular form under high vacuum conditions [52,53,54,55].

However, unlike the magnetron co-sputtering and multi-arc ion plating technologies, the EB-PVD cannot fabricate CSAFs. Therefore, a specific design must be implemented. Accordingly, a rotatable shadow mask composition spread alloy film deposition tool was developed by Andrew [11,56,57]. Fe-Al-Ni ternary composition spread alloy films were deposited using three confocal e-beam evaporators attached to the tool, one for each elemental component. Partial, line-of-sight shadowing by rotationally adjustable masks mounted between each evaporator and the substrate was used to obtain a gradient in the evaporative flux of each element across the substrate surface (Figure 3).

Similar to the multi-arc ion plating technology, the efficiency of the EB-PVD method for film preparation is also high, enabling the production of thick films in a short period. However, the limitations or challenges of EB-PVD include:(a)To achieve a compositionally graded film, new equipment with masks in front of the targets must be designed, which may increase the associated costs.(b)It is difficult to precisely control the composition of the CSAFs.(c)Certain highly saturated vapor pressure elements, such as Nb and Mo, cannot be deposited onto the substrates [58,59,60,61,62,63,64].

### 2.4. Additive Manufacturing

Powders from each hopper are drawn into a gas line of flowing argon by the rotation of an auger located at the base of each hopper. These powders are aggregated in a central gas line and mixed by the turbulent gas flow during transit to the printhead. At the printhead, the mixed powder is sprayed out by the nozzles, with rotational symmetry similar to the optic axis of the printhead, where it encounters a laser impinging on the surface of the build plate, schematically represented in Figure 4.

Upon reaching the printhead, the combined powder is emitted through four nozzles, evenly spaced around the optical axis of the printhead. At this point, the power encounters a laser beam directed at the build plate. The laser creates a molten pool on the build plate, into which the incoming powder, heated by the laser, is incorporated. This molten pool can then be moved across the build plate by adjusting the stage upon which it is located. This process leaves behind solidified material as it advances. As the laser follows the designated build path, a continuous interface is maintained between the solid and liquid states. By adjusting the stage, the laser trajectory across the build plate can be controlled, allowing the material to be deposited in various custom shapes. The quantity of powder (A, B, C, and D in Figure 4) can be controlled by the augers, allowing for the arbitrary variation of the combined powder’s composition. As a result, many parts with different compositions can be produced [65].

Katharine et al. [66] proposed a high-throughput laser deposition method to prepare CSAF (Figure 5). An alloy substrate was prepared using arc melting and casting. The laser and powder stream raster over the surface to create a CSAF. That is, the laser generates a molten pool on the surface of the substrate, into which a continuous flow of alloying powder is directed. The laser and the powder stream move back and forth across the surface to form a single layer, and additional layers can be applied as needed to build a three-dimensional structure. The powder feed rate for each layer can vary. Thus, numerous patches with different compositions can be obtained.

Additive manufacturing technology can produce relatively thick coatings and is convenient for subsequent performance testing of oxidation resistance, mechanical properties, and corrosion resistance. However, the implementation cost of this technology is high, partly due to the high cost of the machine and partly due to the expensive powders required for certain new materials, such as high-entropy alloys [67,68,69,70,71].

### 2.5. Chemical Bath Deposition

Chemical bath deposition is another method suitable for gradient thin film preparation due to its flexible synthetic chemistry, easy-achieved operation requirement, and low cost [72,73,74,75,76,77,78]. Yong Xiang et al. [79] presented a high-throughput combinatorial technique for continuous thin film preparation that relied on chemical bath deposition, as shown in Figure 6. A 3 × 3 discrete mask was placed on the substrate to produce Cu (In, Ga) Se film (CIGS) absorber samples (Figure 6a). Gradient thin films with varied thicknesses can be fabricated by controlling the lifting speed and rotation mode of a substrate and coating the Mo film with the substrate. As shown in Figure 6b, 2/3 of a substrate is immersed into the reaction solution (Na_2_S_2_O_3_ + SbCl_3_). The bottom 1/3 of the substrate is then immersed in the reaction solution. Subsequently, the substrate is rotated 90° clockwise, and 2/3 of the substrate is immersed into the reaction solution. Next, 1/3 of the substrate is lifted with the bottom 1/3 remaining in the solution. A thin film with varied thicknesses is obtained, and the sample is placed into a tube for high-temperature annealing (Figure 6c). Ultimately, a doping CIGS library is obtained. Chemical bath deposition is not widely used in high-throughput preparation as, compared to other high-throughput preparation methods, it is more complex to prepare CSAFs; moreover, the chemical bath deposition can only be used to prepare a few types of CSAFs, which limits its applicability.

In addition to the techniques mentioned above, high-throughput pulse laser ablation (HT-PLA) is a versatile method for the rapid creation of multi-metallic nano-particles that are crystalline and uniform in size and composition, using pure metal powders that are readily available [26].

## 3. High-Throughput Characterization Methods

The combinatorial film’s characteristics include microstructure, oxidation, corrosion tested, and mechanical properties. Due to the low hardness of the single crystalline Si wafer, the film used for scanning electron microscopy (SEM) and transmission electron microscope (TEM) is deposited on the Si substrate. However, the choice of substrate is influenced by many physical factors, including adhesion and the possibility to achieve heteroepitaxy in some cases. It may also be influenced by foreseen applications for thin or thick films. Therefore, the Si wafer is not the only option. Due to weak adhesion between the Si wafer and the alloy films, spallation occurs during oxidation or corrosion tests. Therefore, the film deposited on metal substrates is used for annealing treatment, oxidation, corrosion, and mechanical tests.

### 3.1. Composition and Microstructure

The surface morphology, cross-section morphology, and composition of the CSAFs are examined via SEM with an energy dispersive spectroscopy (EDS) attachment; the composition measurements are spaced by a certain distance, arranged to form composition mapping.

To avoid peak occurrence from substrates, the crystalline structure and the phases of the CSAFs are analyzed via glancing incidence X-ray diffractometry (GIXRD), with a Cu Kα radiation and a scanning rate of 3–8°/min. Moreover, Raman spectroscopy and XPS are used to characterize the microstructure. The Raman spectrum is obtained from oxidized or corroded CSAFs to identify the constituting phases. The distance between each data point is consistent with SEM measurements. XPS depth profiling is employed to determine the cross-sectional composition of the CSAF at selected sites of interest for Raman analysis. The surface roughness of the film is measured by an atomic force microscope (AFM) [64]. Moreover, ellipsometry is a common optical measurement technique that characterizes light reflection (or transmission) from a sample. It is a highly sensitive and nondestructive technique for detecting changes in thin film thickness or refractive index [73,74].

### 3.2. Oxidation and Corrosion Properties

The CSAFs are oxidized in an electric tube furnace in an air environment. To prevent the introduction of impurities during the oxidation experiments, the samples are placed in a ceramic crucible. Notably, the thickness of the CSAFs limits the oxidation conditions that can be reasonably studied. Therefore, it is advisable to conduct short-term or low-temperature oxidation experiments. EDS, XPS, Raman, etc., are used to characterize the CSAF’s oxides. Moreover, a thermogravimetric analyzer (TGA) can measure oxidation mass gain curves.

### 3.3. Mechanical Property

Instrumented nanoindentation can be programmed to perform indentations at multiple locations on a material surface in a single run, making it a valuable tool for conducting high-throughput measurements of the material library mechanical properties. The hardness, modulus, and yield stress of the CSAFs can be obtained using a nano-indenter [65,66]. The distance between each indentation on the film surface should be consistent with the SEM measurements. The indents are arranged to form nanoindentation mapping, including the hardness, modulus, and yield stress.

## 4. High-Throughput Screening Methods

High-throughput screening methods enable researchers to carry out multiple investigations within a relatively short timeframe. This is typically achieved through swift sequential automation or concurrent measurements, involving myriad material variables. Consequently, the application of high-throughput experiments serves the purpose of swiftly screening materials; the insights gained from these experiments serve as a foundation for subsequent, more detailed investigations.

### 4.1. Oxidation Resistance Screening

High-throughput approaches have been developed to accelerate the process of screening the oxidation-resistant composition of alloys [39,45,80,81,82,83]. The main steps of high-throughput screening for oxidation properties are as follows: (1) Analyzing and testing the types and distribution of oxides; (2) Plotting a mapping of the oxides (types and distribution); (3) Comparing the mapping of the CSAFs to analyze the alloy composition corresponding to each type of oxide.

Nb, Cr, Ti, and Si compositions were measured using EDS spectra at 121 discrete points on the Nb-Si-based alloy film. Atomic compositions in real space on the CSAF are shown in Figure 7, and the precise coordinates can be obtained from the x-axis and y-axis values. Short-term (50 min) oxidation behavior at 1250 °C was characterized across the oxidized film. The film exhibits three regions of distinct oxidation behavior: CrNbO_4_, CrNbO_4_ + Nb_2_O_5_ + TiNb_2_O_7_, and Nb_2_O_5_ + TiNb_2_O_7_ (Figure 8). According to the CSAF composition analysis (Figure 7), the composition for establishing a protective CrNbO_4_ scale is determined.

### 4.2. Corrosion Resistance Screening

The screening process for the corrosion resistance of alloys also includes the three steps for assessing oxidation properties, with slight variations in the methods used to characterize the anti-corrosion performance of thin films.

Gao et al. [4] fabricated nanocrystalline Fe-Cr-Ni films (22 nanocrystalline) with a thickness of ~5 μm by magnetron co-sputtering technology. The potentiodynamic polarization test can be applied to characterize the corrosion properties. The film’s breakdown potential (E_b_) distribution map is then generated, and the characteristic regions are plotted in the Schaeffler structure diagram (Figure 9). Considering the Schaeffler structure diagram, Figure 10 shows that the four alloys (S1, S2, S3, and S4) had a higher α phase proportion. The Fe−Cr−Ni alloy films covered 10.65−28.36 wt. % Cr and 7.47−24.57 wt. % Ni. Furthermore, typical phases (i.e., ferrite (α) and austenite (γ)) were formed in the film. The optimal region with excellent corrosion resistance was quickly demarcated in a wide composition range.

In addition to conventional materials, novel materials, such as high-entropy alloys, can be screened using high-throughput techniques. Peter K. Liaw et al. [84] successfully utilized magnetron co-sputtering to achieve a combinatorial material library of Alx(CoCrFeNi)100-x (3d transition element HEA), covering a range of 4.5–40 atomic percent Al. The corrosion properties of the combinatorial samples were evaluated through electrochemical tests after being immersed in a 3.5 wt.% NaCl solution. Complementary analysis using X-ray photoelectron spectroscopy revealed variations in the composition of the passivated films that formed on the sample surface after immersion.

### 4.3. Mechanical Property Characterization

The main steps of high-throughput screening for mechanical properties are as follows: (1) Test the properties (hardness, modulus, etc.) of the CSAFs at different locations; (2) Plot a map of the results; (3) Compare the mappings of the CSAFs to analyze the alloy composition corresponding to each location.

Nanoindentation tests can provide reliable information on mechanical properties, such as elastic modulus, yield strength dislocation nucleation stress, and hardness. Therefore, a high-throughput testing method via nanoindentation mapping can be conducted to evaluate the mechanical properties of materials.

Hardness is a main property of alloys. Yong Zhang et al. [80] fabricated gradient composition WTaCrFeNi refractory high-entropy alloy (RHEA) composition gradient films using a three-target (W, Ta, CrFeNi) magnetron co-sputtering technique. Figure 11 shows the size of the targets, the magnetron-sputtering process, and the W-Ta-CrFeNi composition gradient films. The films exhibited BCC (near W or Ta target) and amorphous (near CrFeNi target) structures. The W_15.39_Ta_38.81_Cr_14.58_Fe_15.45_Ni_15.77_ alloy film exhibits the maximum hardness of ~20.6 GPa.

Zhang et al. [80] prepared gradient composition films of W-Ta-CrFeNi high-entropy alloy; hardness and elastic modulus mapping of the films were then obtained with a nano-indenter (Figure 12), from which the hardness and elastic modulus values were obtained. The region near the W and Ta targets exhibits a higher hardness than the center of the three-dimensional diagram region (Figure 12a). However, the elastic modulus mapping indicates a higher elastic modulus region near the W-rich region (Figure 12b). Combined with the microstructure mappings, the relationship between mechanical properties and the composition and phase structure could be acquired. A novel nanoindentation mapping technique was employed by Tong et al. [82] to correlate the microstructure, composition, and mechanical properties of CoCrNiMox (x = 0.2, 0.4, 0.6, 0.8) high-entropy alloys. The nanoindentation pattern was 25 × 25, and the minimum spacing between two indents was 2 μm (Figure 13a–d). An interesting observation is the comparable Young’s modulus and the hardness among the microstructure (Figure 13e). From the results, the correlation between microstructure, composition, and mechanical properties with spanning Mo content was quantified at the microscale. 

In addition to hardness, modulus is a performance indicator. Sn-Zn-xCu (7 ≤ × ≤ 20.2) CSAFs were fabricated (15 samples with a 3 mm interval), and the mechanical properties of the films were investigated [85]. Two targets were used in this work: Cu concentration range from ~20.2 to 2.3 wt.%, and Sn content from 74.4 to 92.8 wt.% (Figure 14a,b). The Young’s modulus of the sample was measured by an in-situ nanomechanical testing system. With the decreased Cu content, the samples were composed of matrix Sn, Cu_6_Zn_5_, and Cu_5_Zn_8_; the Young’s modulus of the films increased (Figure 14c,d).

High throughput magnetic characterization was used to optimize the composition of L1_0_-based FePt thin films [86,87]. Fe-Pt films were fabricated by magnetron sputtering of a Fe target partially covered by Pt foils (Figure 15a) [88]. The composition was varied by altering the size and position of the Pt foils. A scanning polar Magneto-Optic Kerr effect system was developed to perform magnetic characterization. The coercivity reached ~1.5 T in the 55–60 at.% Pt range.

## 5. Current Challenges and Future Outlook

The traditional trial and error method for compositional screening is greatly limited. To overcome this challenge, high-throughput methods are commonly used to generate combinatorial material libraries, particularly for complex, concentrated alloys like high-entropy alloys and superalloys, which have a relatively infinite number of possible combinations in their multi-dimensional compositional space.

To date, many studies have been conducted on combinatorial film high-throughput methods. However, certain challenges remain. Therefore, further work is needed to broaden the knowledge of these new techniques. In this section, various ideas are introduced to provide possible lines of research to improve these methods and facilitate innovative designs.

### 5.1. Current Challenges

Despite the high-throughput methods’ advantages for advanced materials, the following challenges must be addressed to realize their potential fully. 

(a) The composition and distribution of the microstructure within the combinatorial films must be precisely regulated. Properties of the alloys, such as oxidation, corrosion, mechanical strength, ductility, etc., are determined by the composition and microstructure. To comprehensively and accurately optimize the performance of alloys, the CSAFs prepared via high-throughput methods must contain more microstructural features. Furthermore, the film’s microstructure can be precisely controlled, allowing the desired microstructure to be obtained. Thus, further research is needed to study the influence of preparation technology on the composition and microstructure of the CSAFs.

(b) The post-treatment process requires further investigation. Due to the excellent glass-forming ability of the system, comprising high melting point elements and a high cooling rate during the deposition process, amorphous phases can be easily observed in the film. Crystallization of the film occurs after annealing treatment, while the interdiffusion between the combinatorial film and the substrate cannot be ignored. Moreover, the composition range after annealing treatment may differ from the as-deposited film; thus, reasonable heat treatment procedures should be studied further.

(c) The intrinsic relationship associated with using the thin film to characterize the properties of bulk materials must be established. Experimental constraints associated with the rotatable shadow mask (RSM) CSAF deposition tool limit the thickness of the CSAF that can be created, consequently limiting the oxidation temperature/time that can be reasonably studied. Concerning practical applications, oxidation at lower temperatures is a very early/transient behavior compared to that typically considered for alumina formers, where bulk alloys may be oxidized in environments at 600 °C to 1200 °C for hundreds or thousands of hours [56]. 

The degree to which the behavior of individual sites on the CSAFs is representative of that of bulk alloys with similar compositions under identical environmental conditions is yet to be determined, even when the investigation is restricted to the initial oxidation phase. One major reason for this is that there are likely significant differences in the microstructure of thick alloy films (columnar crystal, nanocrystalline, or amorphous) compared to bulk alloys; these differences might affect the oxidation process. Moreover, the CSAFs might oxidize differently from bulk alloys if sufficient oxide formation occurs such that one or more metallic species are significantly depleted from the film, causing a change in its effective composition and affecting subsequent oxidation. Additionally, mechanical properties like tensile strength, compression strength, etc., tested by dog-bone flat samples, cylindrical tensile specimens, etc., cannot be obtained by thin film.

(d) High-throughput preparation and characterization equipment must be designed. Currently, conventional equipment, such as magnetron sputtering, multi-arc ion plating, and electroplating, is used to prepare CSAFs. These devices were not originally intended for the preparation of the CSAFs. Therefore, their use in high-throughput preparation will lead to the following issues: (1) the prepared film thickness is too thin; (2) the range of film composition is difficult to control; (3) the preparation cycle is lengthy; and (4) the production cost is high.

Moreover, conventional methods such as SEM, TEM, Raman, etc., remain the primary methods used to characterize the CSAFs. These analyses are resource-intensive and time-consuming. Therefore, advanced high-throughput preparation and characterization methods should be introduced to accelerate the development of combinatorial film high-throughput techniques.

(e) Techniques or theories must be developed to characterize the mechanical properties of bulk materials using CSAFs. The mechanical properties of bulk materials include hardness, tensile strength, compressive strength, impact toughness, creep resistance, etc. These properties are closely related to the microstructural of the bulk materials, such as phase composition, grain sizes, and crystallographic orientation, which are affected by the preparation and processing techniques. Due to the limitations of thin films, it is challenging to perform tests on most of the mechanical properties of bulk materials, and it is not possible to employ the same preparation and post-processing techniques as those used for bulk materials.

### 5.2. Future Outlook

#### 5.2.1. High-Throughput Synthesize Apparatus

In the future, a unique apparatus must be designed to satisfy the high-throughput synthesis demands and improve the level of combinatorial film throughput. Hong Wang et al. [89] created continuous thickness gradients across a substrate using a moving shutter (Figure 16a). The substrate was rotated by 120° after deposition of each element (Cr, Co, Cu). The total thickness of the multilayer thin-film at all locations of the combinatorial material chip was 100 nm for all film stacks, obtained with a modulation period of 100 nm, 50 nm, 20 nm, and 10 nm, and the corresponding number of repetitions, respectively (Figure 16b). Based on the thickness ratio among the individual nanoscale monolayers (Cu, Cr, Co), the resulting stoichiometry covered the entire phase diagram.

#### 5.2.2. High-Throughput Characterization Apparatus

The characterization methods discussed above are conventional testing methods that require significant time and effort. Therefore, they can be combined with high-throughput techniques to achieve more efficient sample handling and data analysis. For instance, automated sample slicing and loading systems can be employed to expedite the processing of multiple samples. In contrast, parallel data acquisition and processing methods can accelerate the speed of data retrieval and analysis.

#### 5.2.3. Relationship between the Properties of Thin Films and Bulk Materials

The intrinsic relationship between using thin film techniques and characterizing properties in bulk materials should be further developed. A high-throughput preparation method is employed to synthesize 85 combinatorial alloys in a 13-principal element (Co, Cr, Fe, Ni, Mn, Cu, Ti, Nb, Ta, Mo, W, Al, and Si) alloying space of Cantor alloys [3]. As shown in Figure 17a,b, the VEC (valence electron concentration), Ω (Ω = T_m_S_mix_/|ΔH_mix_|; T_m_ refers to melting temperature, S_mix_ is the entropy of mixing, ΔH_mix_ denotes the enthalpy of mixing), and δ (atomic size difference) determines the phase (IM or SS phases) of high-entropy alloys. The microstructure influences the hardness (Figure 17c). The composition-microstructure-microhardness data offers better evaluation and direct comparison of the alloying effect on structures and properties from the selection or combination of additions. However, limited research has been reported in this area, lacking significant theoretical breakthroughs.

## 6. Conclusions

Traditional trial-and-error methods in materials research and development consume significant resources and time. The concepts of high-throughput preparation and characterization have been introduced to identify promising structural alloys with desired properties. This review article presentes the recent advancements in the gradient thin-film deposition for high-throughput preparation of material libraries regarding high-throughput synthesis, characterization, and screening methods and their associated oxidation, corrosion, and mechanical properties, as well as the current challenges and future outlooks.

(1) The combinatorial film high-throughput synthesis techniques rely on the production of compositional gradients to form a range of alloy compositions, as is the case for magnetron co-sputtering, multi-arc ion plating, e-beam evaporation, electrodeposition, and additive manufacturing. Among these techniques, magnetron co-sputtering has been widely used for high-throughput preparation of composition spread alloy films (CSAFs).

(2) The combinatorial films’ characteristics include microstructure, oxidation, corrosion test, and mechanical properties. The properties screening methods can be summarized as (1) test the properties of the CSAFs at different locations; (2) plot a mapping of the results; (3) compare the mappings of the CSAFs to analyze the alloy composition corresponding to each location.

(3) The current challenges associated with combinatorial film high-throughput techniques include precise control of the film composition and microstructure, further research on post-treatment processes, establishing the intrinsic relationship between thin film characterization and the properties of bulk materials, developing high-throughput equipment for preparation and characterization, as well as techniques or theories for characterizing the mechanical properties of bulk materials.

(4) The further evolution of combinatorial high-throughput film techniques lies in developing advanced high-throughput synthesis and characterization apparatus, as well as studying the relationship between the properties of thin films and bulk materials.

## Figures and Tables

**Figure 1 materials-16-06696-f001:**
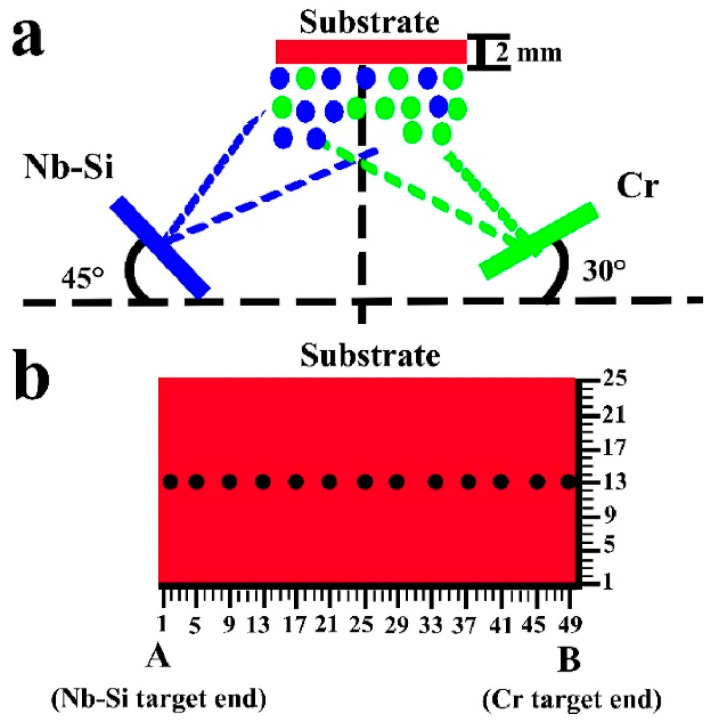
Schematic image of (**a**) magnetron sputtering for generating composition spread alloy film (**b**) compositionally varied Nb-Si-based alloy film [39].

**Figure 2 materials-16-06696-f002:**
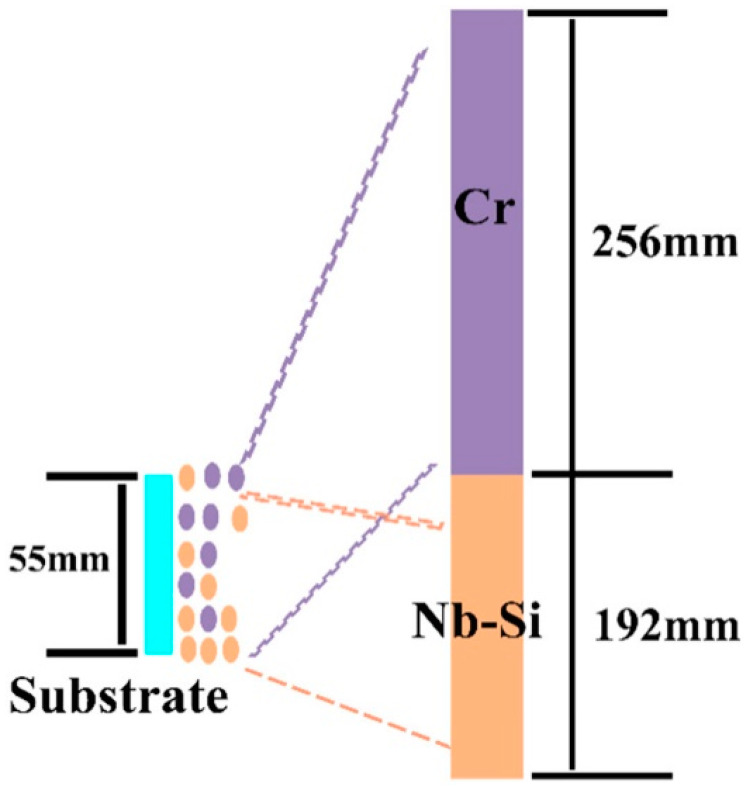
Schematic image of multi-arc ion plating for generating alloy film [45].

**Figure 3 materials-16-06696-f003:**
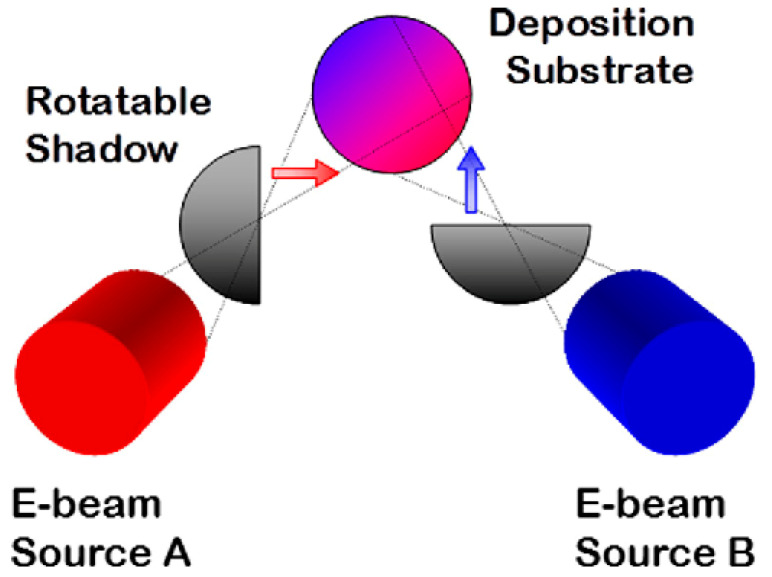
Conceptual schematic of the rotating shadow mask film deposition tool [57].

**Figure 4 materials-16-06696-f004:**
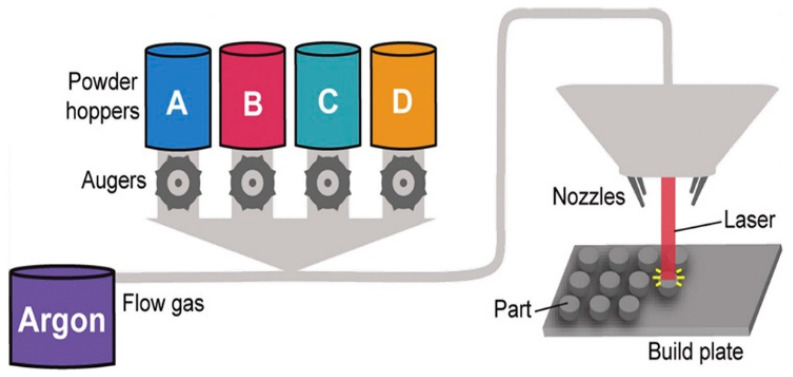
Schematic illustration of additive manufacturing high-throughput method. Each hopper (A–D) is filled with a single elemental powder [65].

**Figure 5 materials-16-06696-f005:**
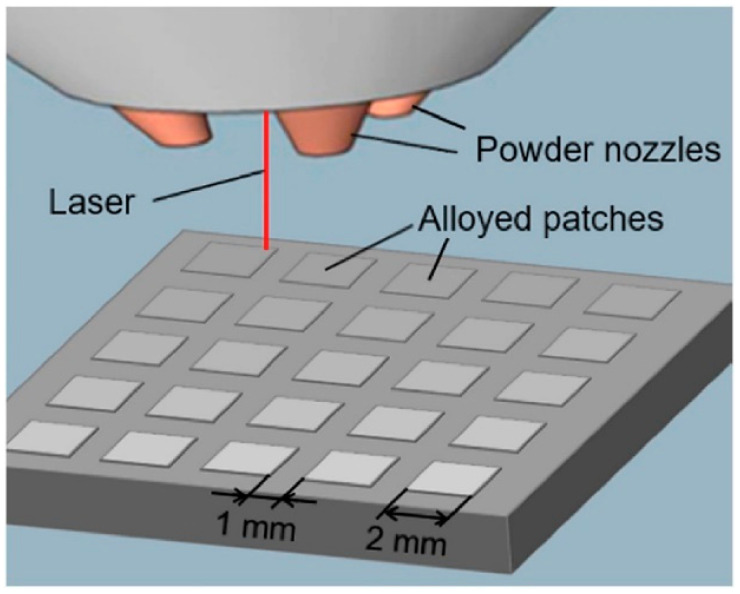
Schematic of the laser deposition technique used to create the alloy library [66].

**Figure 6 materials-16-06696-f006:**
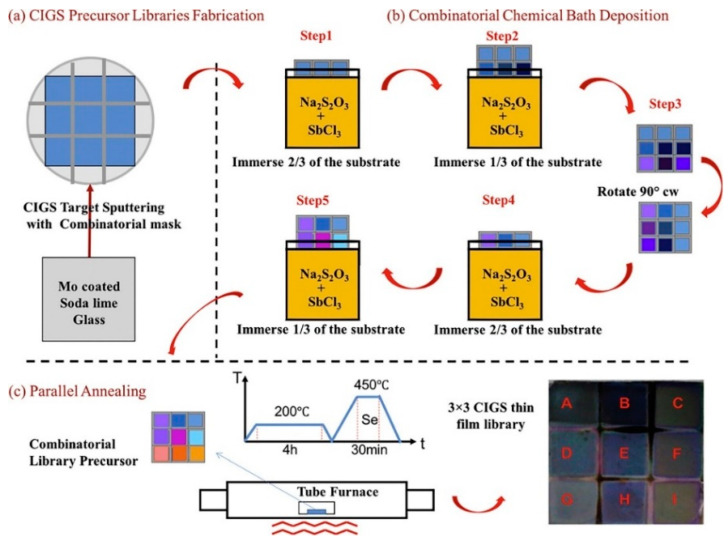
Scheme of combinatorial chemical bath deposition of Cu(In, Ga)Se thin film libraries with variation thickness. (**a**) CIGS precursor libraries fabrication, (**b**) combinatorial chemical bath deposition, (**c**) parallel annealing [79].

**Figure 7 materials-16-06696-f007:**
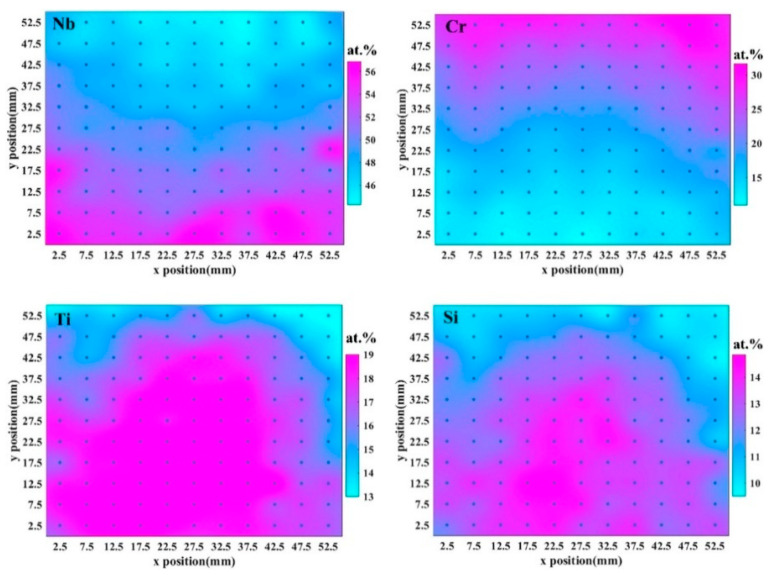
Atomic compositions in real space on Nb-Si-based CSAF [31].

**Figure 8 materials-16-06696-f008:**
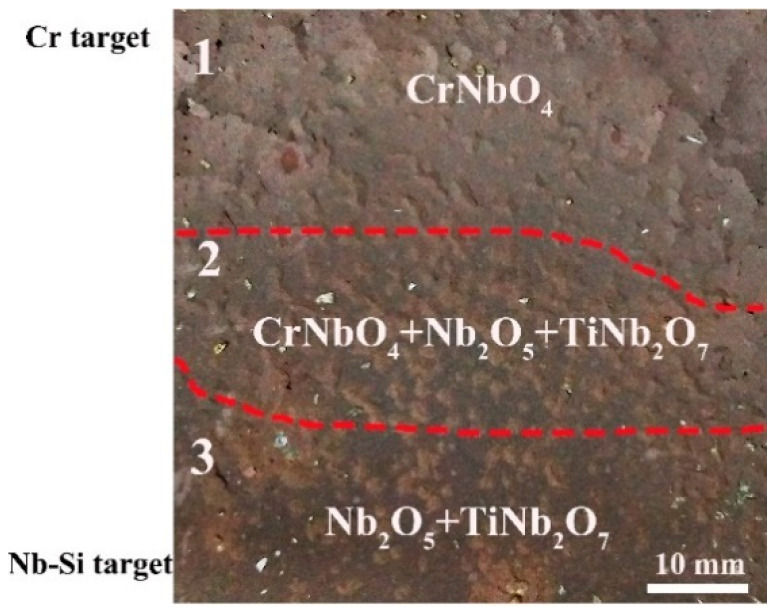
Distribution of oxide phases across on Nb-Si based CSAF. (1) area of CrNbO_4_; (2) area of CrNbO_4_ + Nb_2_O_5_ + TiNb_2_O_7_; (3) area of Nb_2_O_5_ + TiNb_2_O_7_ [45].

**Figure 9 materials-16-06696-f009:**
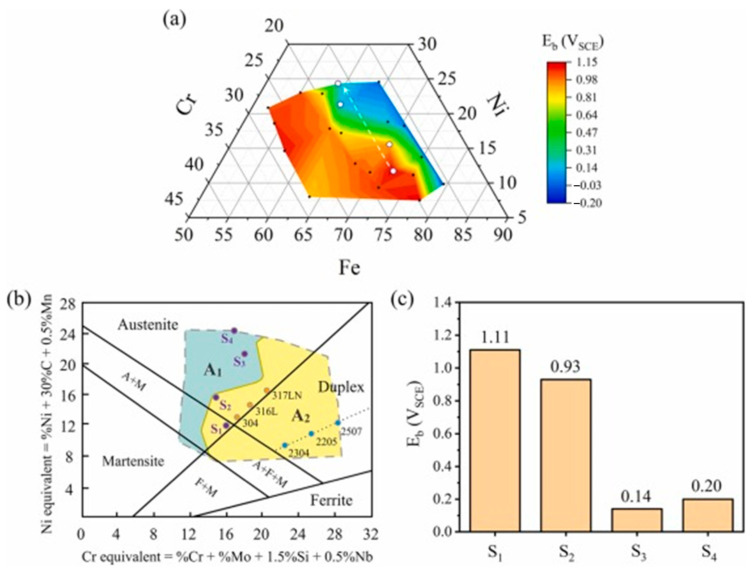
(**a**) E_b_ distribution map of nanocrystalline Fe-Cr-Ni films. (**b**) Schematic diagram of the E_b_ distribution characteristics based on the Schaeffler structure diagram. (**c**) The E_b_ of S_1_, S_2_, S_3_, and S_4_ [4].

**Figure 10 materials-16-06696-f010:**
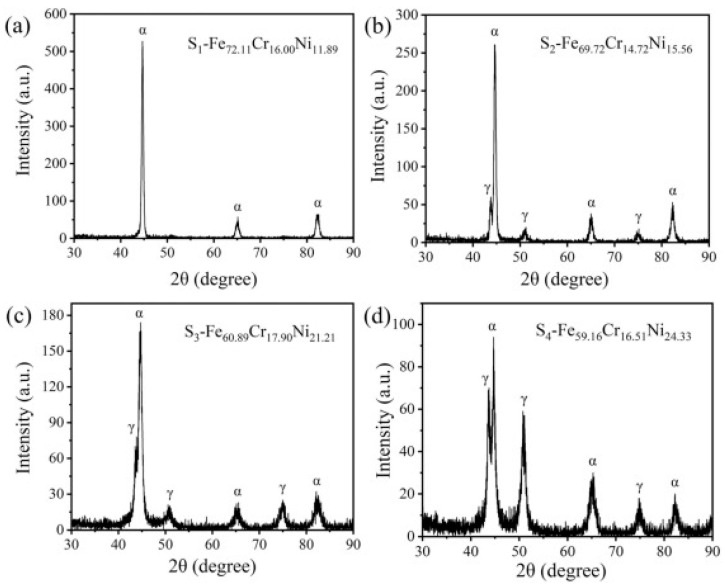
XRD patterns of (**a**) Fe_72.11_Cr_16.00_Ni_11.89_ (S_1_), (**b**) Fe_69.72_Cr_14.72_Ni_15.56_ (S_2_), (**c**) Fe_60.89_Cr_17.90_Ni_21.21_ (S_3_), and (**d**) Fe_59.16_Cr_16.51_Ni_24.33_ (S_4_) [4].

**Figure 11 materials-16-06696-f011:**
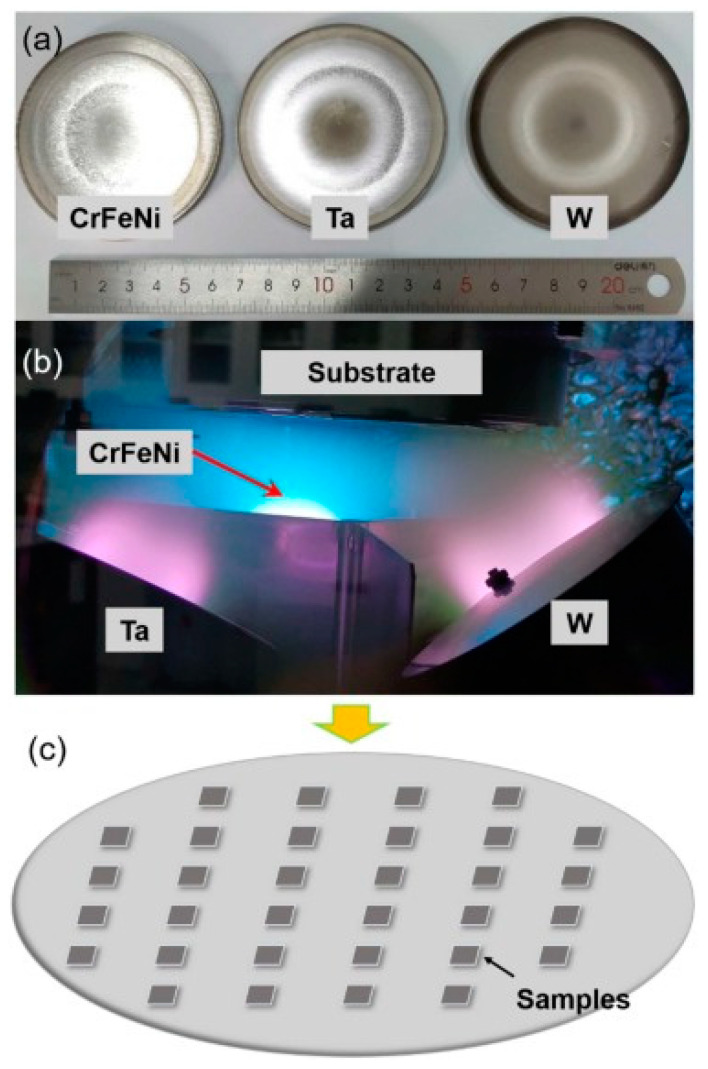
(**a**) Targets for magnetron sputtering, (**b**) magnetron-sputtering process, and (**c**) W-Ta-CrFeNi composition gradient films. Using the process illustrated in Figures A and B, the thin film depicted in Figure C can be produced. [80].

**Figure 12 materials-16-06696-f012:**
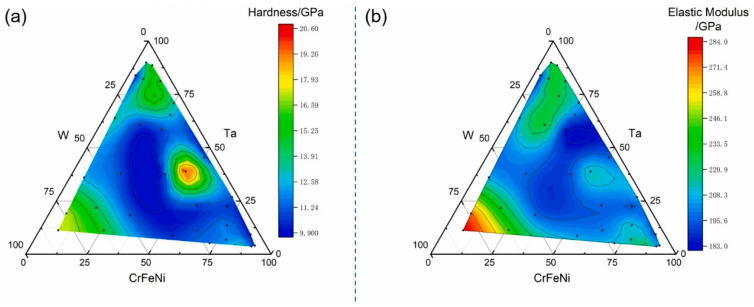
(**a**) Hardness and (**b**) elastic modulus of W-Ta-CrFeNi gradient thin film [80].

**Figure 13 materials-16-06696-f013:**
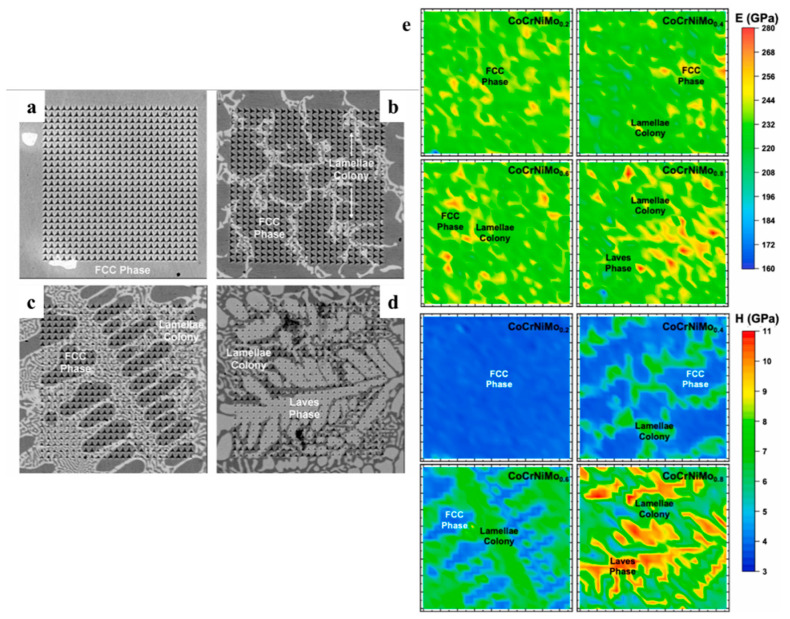
(**a**–**d**) Microstructures of the indented regime in CoCrNiMox films, (**e**) Young’s modulus and hardness maps of the selected regime [82].

**Figure 14 materials-16-06696-f014:**
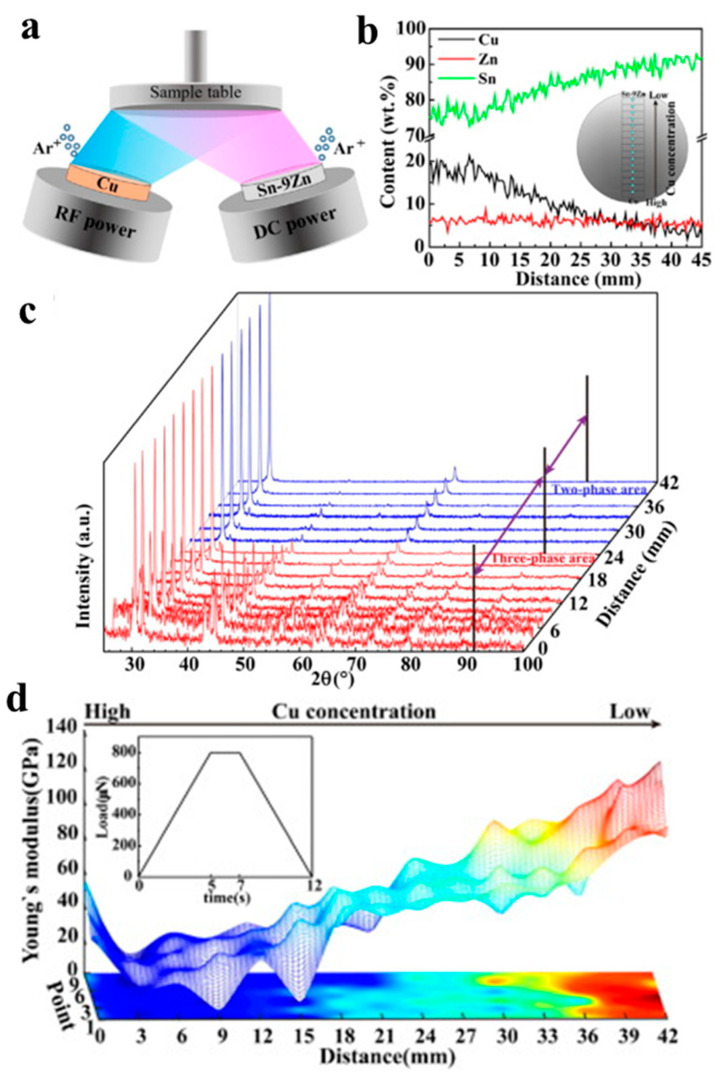
(**a**) Schematic of the preparation method; (**b**) composition gradient of the films; (**c**) XRD patterns of the films; (**d**) Young’s modulus of the Sn-Zn-Cu thin film can be fitted by Matlab (R2016a) [69].

**Figure 15 materials-16-06696-f015:**
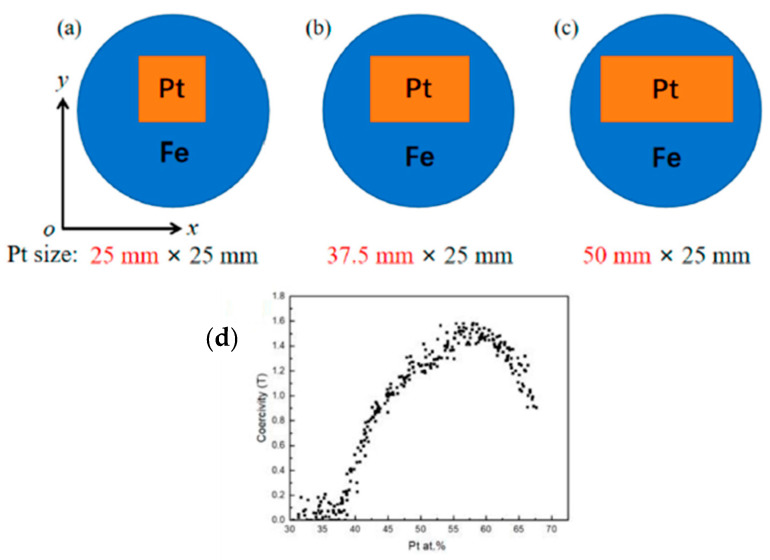
(**a**–**c**) Schematic diagrams of the targets used to produce compositionally graded Fe-Pt films; (**d**) plot of coercivity as a function of Pt content [88].

**Figure 16 materials-16-06696-f016:**
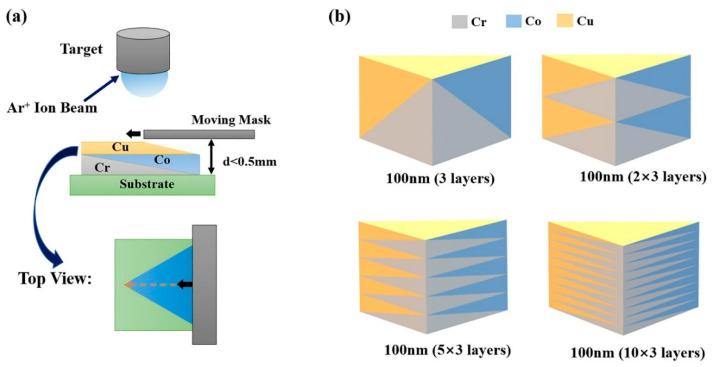
Schematic diagram of combinatorial multilayer thin films. (**a**) Deposition procedure with a moving mask; (**b**) cross-section of the sample with different modulation periods [89].

**Figure 17 materials-16-06696-f017:**
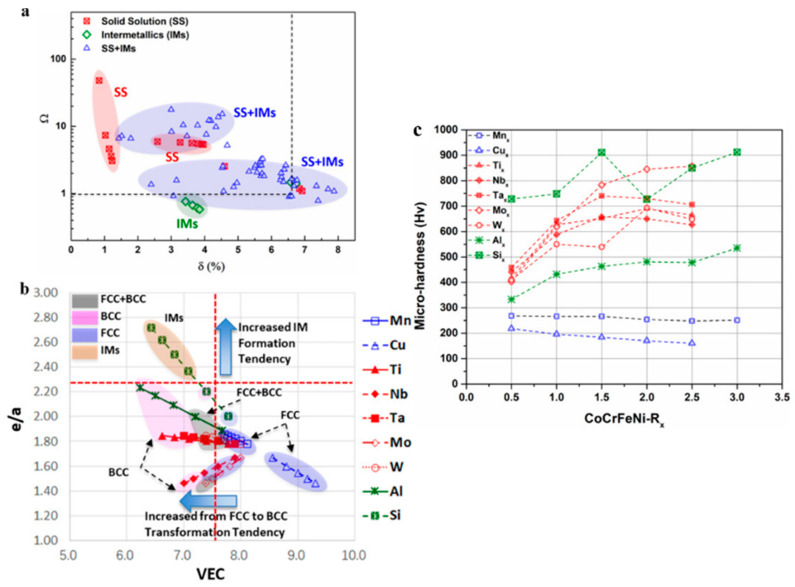
(**a**) Phase formation determined by Ω and δ; (**b**) Phase classification determined by VEC and e/a; (**c**) The microhardness variation with the CoCrFeNiRx (x = 0.5–3.0) [3].

## Data Availability

Not applicable.

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
