# Peer review of "A Comprehensive Review on Combinatorial Film via High-Throughput Techniques"

_materials, 2023, doi:10.3390/ma16206696_

Round 1

Reviewer 1 Report

Journal: Materials

Manuscript Number: materials-2610943  
Title: A comprehensive review on combinatorial film via high-throughput techniques

In this paper, the authors summarized recent development progress in gradient thin-film deposition fabricated by the magnetron sputtering, multi-arc ion plating, e-beam evaporation, additive manufacturing and chemical bath deposition reported so far, and provides readers with a fundamental understanding of this research field.

The quality of the paper is good. It is technically sound. The results seem promising. It is written in good language.  It can be published in Materials. I have the following comments.    

My comments:

1-      As a review paper, more previous studies should be added to the introduction section.

2-      Figures 9 and 10 can be enlarged.

3-      References are written in different formats. For example, see references 72 and 73.  Please be consistent.

4-      The motivation behind this work is not clearly presented. Please state and explain.

5-      How can this work help add to the field.

6-      Ellipsometry is a significant technique for the characterization of thin films. This has to be mentioned in the introduction section.  It has many applications. You can use the following citations:  Thin solid films, vol. 518, No. 19, 5610-5614, 2010. doi:10.1016/j.tsf.2010.04.067   

Physica Scripta, Vol. 83, No. 2, 7 pages, 025701, (2011). doi:10.1088/0031-8949/83/02/025701

Author Response

Comments 1: As a review paper, more previous studies should be added to the introduction section.

Response 1: Thank you for pointing this out. More previous studies have been added to the introduction section.

Comments 2:   Figures 9 and 10 can be enlarged.

Response 2: Thank you for pointing this out. Figures 9 and 10 have been enlarged.

Comments 3:   References are written in different formats. For example, see references 72 and 73. Please be consistent.

Response 3: Agree. The references have been standardized, and the revised areas have been highlighted.

Comments 4: The motivation behind this work is not clearly presented. Please state and explain.

Response 4: The conventional trial-and-error approaches for materials research and development are resource and time-consuming. It becomes imperative to develop a method that is efficient and systematic for the study of structural alloys with targeted properties. The establishment of the Materials Genome Initiative (MGI) has become a turning point in data-driven materials science, and the database has gradually evolved into a data center that provides materials data and fundamental analysis services.

Composition spread alloy films (CSAFs), first invented in the field of chemistry, are one of the most mature and widely used technologies for high-throughput preparation of material libraries. Many useful discoveries in the field of materials science have been made as a result of adopting CSAFs approaches, particularly in the areas of corrosion, oxidation, magnetic, and mechanical properties.

Several review articles have focused on the MGI, providing valuable information about the current situation and development of the MGI. However, there are limited reviews dedicated to the comprehensive presentation of fabrication, screen methods, micro-structures, properties, and challenges of the CSAFs approaches. Therefore, this review focuses on the application of combinatorial film high-throughput preparation and characterization techniques. The progress, and application fields, as well as common problems of this specific topic, are within the scope of the review.

The state and explanation have been added in the introduction section and highlighted.

Comments 5:  How can this work help add to the field.

Response 5: High-throughput strategies offer an alternative to one-by-one or single-step experiments by allowing researchers to perform multiple experiments in a relatively short time. The composition spread alloy films (CSAFs) is one of the most mature and widely used technology for high-throughput preparation of material libraries. Many useful discoveries in the field of materials science have been made as a result of adopting CSAFs approaches, particularly in the areas of corrosion, oxidation, magnetic, and mechanical properties.

    Several review articles have focused on the materials genome initiative (MGI), providing valuable information about the current situation and development of the MGI. However, there are limited reviews dedicated to the comprehensive presentation of fabrication, screen methods, micro-structures, properties, and challenges of the CSAFs approaches. Therefore, this review focuses on the application of combinatorial film high-throughput preparation and characterization techniques. The progress, and application fields, as well as common problems of this specific topic, are within the scope of the review.

The state and explanation have been added in the introduction section and highlighted.

Comments 6:  Ellipsometry is a significant technique for the characterization of thin films. This has to be mentioned in the introduction section.  It has many applications. You can use the following citations:  Thin solid films, vol. 518, No. 19, 5610-5614, 2010. doi:10.1016/j.tsf.2010.04.067   Physica Scripta, Vol. 83, No. 2, 7 pages, 025701, (2011). doi:10.1088/0031-8949/83/02/025701

Comments 6: The citations mentioned above have been added in the Introduction section and section 3.1.

Reviewer 2 Report

The manuscript is intended as a review on the field of combinatorial film (metals are mostly considered as components), with a claimed special emphasis on high-throughput techniques for both synthesis and diagnostics.

Combinatorial film production conceptually overlaps with alloying, a fundamental topic which has been certainly treated in a huge number of scientific papers and textbooks. Therefore, it is not straightforward to identify original aspects, or even strong motivations, for this review. Nonetheless, the ties with material libraries, organized in a similar fashion as in genomics, are sound and relatively not much covered in the past literature. Within this frame, Section 5 of the manuscript appears interesting, since it clearly outlines a few scientific and technical issues to be resolved in order for the relevant topic to progress.

This is the main reason why, in my opinion, the paper can be considered for publication.

However, prior to acceptance, Authors must account for several criticisms from my side, as in the following list.

1.     As already mentioned, literature on the general topic of (metal) alloys, including high entropy systems, is practically unlimited. Authors must clearly declare in the manuscript that examples are considered in their review. This applies in particular to both mentioned fabrication methods, where alternative possible approaches (e.g., laser ablation) are neglected, and investigation tools, restricted to some standard, while appropriate, techniques. In particular, emphasis on the analysis of oxidation seems to be relevant only for certain classes of alloys, suffering from high reactivity with ambient oxygen.

2.     The same comment above can be used also for the discussion on the substrate used for deposition, as mentioned at lines 190 and following. I think that choice of the substrate is influenced by very many physical motivations, including adhesion, possibility to achieve heteroepitaxy in some cases, maybe even the presence of foreseen applications for the thin, or thick, film as such. Si is frequently used for the matter of convenience, but I would never define it as the material of choice. 

3.     I noticed that many figures (starting from Fig. 8, with the exception of Fig. 15(a)) are not mentioned in the text. This is not acceptable for a scientific publication and must be carefully amended. Moreover, in some cases the presented figures should be introduced in the text along with some suitable comment and/or an even minimal discussion. For instance, the occurrence of multiple phases in Fig. 8, the occurrence of different crystal phases as a function of the content, seen in the XRD plots of Fig. 9, Fig. 10, 12, 13, 14 in the whole (they are composed by different panels, hence not very straightforward to understand) deserve, in my opinion, a few words in the text aimed at pointing out the scientific message(s) they convey. When revising figures, Authors are also suggested to revise the corresponding captions: for instance, caption to Fig. 11 is not informative at all.

4.     Definition of all quantities mentioned in the paper would also require some additional care. Some of them are in fact not defined, for instance: Eb at line 254 and 262, Omega and delta in the caption to Fig. 17. 

5.     Not all acronyms are defined (spelled) in the text, for instance RSM at line 359.

6.     Style and language can be improved. My feeling is that Sections 3 and 4, in particular, require some more efforts to check syntax and grammar.

Several parts of the manuscript (notably Secs. 3 and 4) require grammar or syntax corrections. Polishing should be applied also to the other parts.

Author Response

Comments 1: As already mentioned, literature on the general topic of (metal) alloys, including high entropy systems, is practically unlimited. Authors must clearly declare in the manuscript that examples are considered in their review. This applies in particular to both mentioned fabrication methods, where alternative possible approaches (e.g., laser ablation) are neglected, and investigation tools, restricted to some standard, while appropriate, techniques. In particular, emphasis on the analysis of oxidation seems to be relevant only for certain classes of alloys, suffering from high reactivity with ambient oxygen.

Response 1: Thank you for pointing this out. We have declared in the manuscript that examples are considered in their review. The literature on high-entropy alloys (HEA) is vast. However, there are only two types of HEAS: 3d transition element HEA and refractory element HEA, and our work involves the introduction of these two types of materials (sections 4.2 and 4.3).

The essence of high-throughput synthesize methods lies in the creation of composition spread alloy films (CSAFs). The preparation methods discussed in this paper are well-developed and extensively employed. Moreover, the introduction of laser ablation has also been added to the manuscript.

The investigation tools introduced in the manuscript are the most typical methods in the field of high-throughput characterization, which can accomplish high-throughput characterization and screening. In addition to the investigation tools mentioned in the manuscript, more characterization methods, such as ellipsometry, have been added.

In addition, the study of oxidation and corrosion performance is aimed at alloys subjected to long-term or harsh environmental conditions. These materials need to withstand such degradation over an extended period of time, and high-throughput techniques enable rapid analysis, thereby saving a significant amount of time and effort.

The revisions have been highlighted in the introduction (the last paragraph) and sections 2.5 (the second paragraph) and 3.1(the second paragraph).

Comments 2:   The same comment above can be used also for the discussion on the substrate used for deposition, as mentioned at lines 190 and following. I think that choice of the substrate is influenced by very many physical motivations, including adhesion, possibility to achieve heteroepitaxy in some cases, maybe even the presence of foreseen applications for the thin, or thick, film as such. Si is frequently used for the matter of convenience, but I would never define it as the material of choice.

Response 2: Thank you for pointing this out. Because of the low hardness of the single crystalline Si wafer, the film used for scanning-electron microscopy (SEM) and transmission electron microscope (TEM) is deposited on the Si substrate. However, the choice of substrate is influenced by many physical motivations, Si wafer is just one of the options. Therefore, the expression used in the text regarding the selection of the matrix is indeed inaccurate. We have revised the expression, and the state has been added in 3. High-throughput characterization methods, and the state has been highlighted.

Comments 3:   I noticed that many figures (starting from Fig. 8, with the exception of Fig. 15(a)) are not mentioned in the text. This is not acceptable for a scientific publication and must be carefully amended. Moreover, in some cases the presented figures should be introduced in the text along with some suitable comment and/or an even minimal discussion. For instance, the occurrence of multiple phases in Fig. 8, the occurrence of different crystal phases as a function of the content, seen in the XRD plots of Fig. 9, Fig. 10, 12, 13, 14 in the whole (they are composed by different panels, hence not very straightforward to understand) deserve, in my opinion, a few words in the text aimed at pointing out the scientific message(s) they convey. When revising figures, Authors are also suggested to revise the corresponding captions: for instance, caption to Fig. 11 is not informative at all.

Response 3: Thank you for pointing this out. We agree with this comment. Therefore, the figures are mentioned in the text, and the presented figures are introduced in the text along with some suitable comment. The revisions have been highlighted.

Comments 4: Definition of all quantities mentioned in the paper would also require some additional care. Some of them are in fact not defined, for instance: Eb at line 254 and 262, Omega and delta in the caption to Fig. 17.

Response 4: Thank you for pointing this out. We agree with this comment. The quantities mentioned in the paper have been defined, and can be found in parts 4.2 and 5.2.3

Comments 5:    Not all acronyms are defined (spelled) in the text, for instance RSM at line 359.

Response 5: Thank you for pointing this out. The acronyms in the paper have been defined. The revisions have been highlighted, and can be found in part 5.1.

Comments 6: Style and language can be improved. My feeling is that Sections 3 and 4, in particular, require some more efforts to check syntax and grammar.

Comments 6: We have checked the text, and some syntax and grammar mistakes have been revised. The revised text has been highlighted, and can be found in sections 3 and 4. 

Reviewer 3 Report

1. For Part 2.1. Magnetron co-sputtering, it briefly mentions the selective loading of elemental materials into sputtering guns in predetermined ratios, but it could benefit from providing more specific information on how these ratios are determined or controlled. This would enhance the reader's understanding of the process.

In the meanwhile, this part does not provide information on the significance or applications of compositionally varied alloy films prepared using magnetron co-sputtering. Adding a sentence or two on the importance of this technique in materials science or industry would provide context for readers.

2. For Part 2.2. Multi-arc ion plating, while it describes the basic setup of multi-arc ion plating, it lacks specific details about how the compositional gradients are achieved. Providing some information about how the two targets with different compositions interact with the substrates to produce gradients would enhance understanding.

3. The description of EB-PVD and the use of shadow masks appears to be technically accurate. However, it would be beneficial to mention any limitations or challenges associated with EB-PVD if relevant.

4. AM can produce relatively thick coatings for performance testing, but it lacks specific details about the AM processes used, such as 3D printing or laser sintering. More information on the specific techniques should be provided.

5. More details about the characterization methods in composition, microstructure, oxidation/corrosion properties, and mechanical properties should be added to enrich the content, such as TGA, hardness, tensile strength…

Author Response

Comments 1: For Part 2.1. Magnetron co-sputtering, it briefly mentions the selective loading of elemental materials into sputtering guns in predetermined ratios, but it could benefit from providing more specific information on how these ratios are determined or controlled. This would enhance the reader's understanding of the process.

In the meanwhile, this part does not provide information on the significance or applications of compositionally varied alloy films prepared using magnetron co-sputtering. Adding a sentence or two on the importance of this technique in materials science or industry would provide context for readers.

Response 1: Thank you for pointing this out. The composition range of the film can be controlled by adjusting the target-substrate angle and target power. This means that the content of elements increases as power increases or as the target-substrate angle changes in the direction of the substrate.

Meanwhile, magnetron co-sputtering has been widely used for high-throughput preparation, which can be used to prepare composition spread alloy films (CSAFs) of various materials. In addition, the equipment can be easily acquired, thereby facilitating the rapid proliferation of its extensive utilization.

The state has been added in section 2.1.

Comments 2: For Part 2.2. Multi-arc ion plating, while it describes the basic setup of multi-arc ion plating, it lacks specific details about how the compositional gradients are achieved. Providing some information about how the two targets with different compositions interact with the substrates to produce gradients would enhance understanding.

Response 2: The Nb-Si based target was placed in the upper part of the target holder and the pure chromium target was placed in the lower part of it, and the substrates were placed in front of the targets. The atoms of the target can be deposited onto the substrate at high temperatures. From the Nb-Si based alloy target end to the Cr target end, the concentration of Cr gradually increases, while the concentrations of Nb, Si, and Ti gradually decrease.

The explanation has been added in section 2.2.

Comments 3: The description of EB-PVD and the use of shadow masks appears to be technically accurate. However, it would be beneficial to mention any limitations or challenges associated with EB-PVD if relevant.

Response 3: Thank you for pointing this out. The limitations or challenges of EB-PVD can be listed as:

(a) To achieve a compositionally graded film, a new equipment with masks in front of the targets needs to be designed, which may increase the cost of experiments.

(b) It is difficult to precisely control the composition of the CSAFs.

(c) Additionally, some highly saturated vapor pressure elements such as Nb and Mo cannot be deposited onto the substrates.

  The statement can be found in section 2.3.

Comments 4: AM can produce relatively thick coatings for performance testing, but it lacks specific details about the AM processes used, such as 3D printing or laser sintering. More information on the specific techniques should be provided.

Response 4: The details of 3D printing can be expressed as: The powders from each hopper are pulled into a gas conduit filled with argon gas through the rotation of an auger situated at the bottom of each hopper. These powders are mixed within a central gas conduit and are subsequently blended as they travel to the printhead, where they are mixed by the turbulent gas flow in the printhead. Upon reaching the printhead, the combined powder is emitted through four nozzles, evenly spaced around the optical axis of the printhead. At this point, the power encounters a laser beam directed at the build plate. The laser creates a molten pool on the build plate, into which the incoming powder, heated by the laser, is incorporated. This molten pool can then be moved across the build plate by adjusting the stage upon which it is located. This process leaves behind solidified material as it advances. As the laser follows the designated build path, a continuous interface is maintained between the solid and liquid states. By adjusting the stage, the trajectory of the laser across the build plate can be controlled, allowing the material to be deposited in various custom shapes. The quantity of powder (A, B, C and D) can be controlled by the augers, which allows for the arbitrary variation of the combined powder's composition. As a result, a large number of parts with different compositions can be produced.

The process can be summarized as: the laser generates a molten pool on the surface of the substrate, into which a continuous flow of alloying powder is directed. The laser and the powder stream move back and forth across the surface to form a single layer, and additional layers can be applied as needed to build a three-dimensional structure. The powder feed rate for each individual layer can be varied. Thus, a large number of patches with different compositions can be obtained.

The state has been added in section 2.4 and highlighted.

Comments 5: More details about the characterization methods in composition, microstructure, oxidation/corrosion properties, and mechanical properties should be added to enrich the content, such as TGA, hardness, tensile strength.

Response 5: Thank you for pointing this out. The details about characterization method of TGA have been added in section 3.2. The details of hardness and tensile strength have been added in sections 3.3 and 4.3.
